# Electrokinetic convection-enhanced delivery for infusion into the brain from a hydrogel reservoir
Jesus G. Cruz-Garza [1,2,6] ✉, Lokeshwar S. Bhenderu [1,2,3,6] ✉, Khaled M. Taghlabi [1,2], Kendall P. Frazee [1,4], Jaime R. Guerrero [1], Matthew K. Hogan [1,2,5], Frances Humes[1,5], Robert C. Rostomily [1,5], Philip J. Horner [1,2,5] & Amir H. Faraji [1,2,5] ✉

Electrokinetic convection-enhanced delivery (ECED) utilizes an external electric field to drive the delivery of molecules and bioactive substances to local regions of the brain through electroosmosis and electrophoresis, without the need for an applied pressure. We characterize the implementation of ECED to direct a neutrally charged fluorophore (3 kDa) from a doped biocompatible acrylic acid/acrylamide hydrogel placed on the cortical surface. We compare fluorophore infusion profiles using ECED (time = 30 min, current = 50 µA) and diffusion-only control trials, for ex vivo ($N = 18$) and in vivo ($N = 12$) experiments. The linear intensity profile of infusion to the brain is significantly higher in ECED compared to control trials, both for in vivo and ex vivo. The linear distance of infusion, area of infusion, and the displacement of peak fluorescence intensity along the direction of infusion in ECED trials compared to control trials are significantly larger for in vivo trials, but not for ex vivo trials. These results demonstrate the effectiveness of ECED to direct a solute from a surface hydrogel towards inside the brain parenchyma based predominantly on the electroosmotic vector.

There are numerous therapeutic interventions ranging from nanoparticles to recombinant viral vectors being developed to treat central nervous system (CNS) diseases[1]. While these advances show great potential in pre-clinical studies, the successful and effective delivery of therapeutics to the CNS, while maintaining targeted delivery and reducing systemic toxicity, remains a challenge[2–4].

The blood-brain barrier (BBB) is a protective layer made from tight junctions between endothelial cells in the CNS along with astrocyte foot processes, microglia, and pericytes[4,5]. The BBB plays a vital role in the protection of the CNS from infectious and neurotoxic agents by preventing transport across the barrier. However, the same properties of the BBB that prevent the transport of harmful molecules also prevent the transport of therapeutic agents into the brain parenchyma from intravascular systemic circulation. Moreover, the BBB may limit the ability to achieve therapeutic levels of an agent in the brain due to systemic toxicity. There are several methods to circumvent the BBB and allow for the delivery of therapeutics. These methods include focused ultrasound[6–9], which works by introducing microbubbles that disrupt the BBB, osmotic or chemical disruption of the BBB[10], nanoparticle delivery[11–14], electrophoretic drug delivery[15–17], and convection-enhanced delivery[2,18,19].

Convection-enhanced delivery (CED) is a method which uses a stereotactic-based cannula inserted directly into the brain parenchyma, therefore circumventing the BBB[2,20]. A pressure-driven infusion of therapeutics is performed through the cannula to provide direct interstitial delivery[5]. Once this pressure gradient is established in the brain, convective forces distribute the agent to the targeted brain site and to the surrounding tissue. CED has gained traction due to the potential for the local delivery of many types of therapeutic agents to specific parts of the CNS with high concentration using stereotaxy[2,3,13,14].

Despite significant potential, some features of CED limit successful therapeutic application as evidenced by mixed results in Phase I/II/III clinical trials for the treatment of glioblastoma and Parkinson's disease[21–23]. These factors include local volume expansion, low target coverage, inability to control the direction of flow after delivery due to varying tissue permeability, leakage of infusate into CSF or other potential spaces in the brain, and backflow in the infusion cannula tract[22–25]. Among these, the pressure gradient required for sufficient local infusions is arguably the most profound

[1]Department of Neurosurgery, Houston Methodist Research Institute, Houston, TX, USA. [2]Center for Neural Systems Restoration, Houston Methodist Research Institute, Houston, TX, USA. [3]Texas A&M University College of Medicine, Houston, TX, USA. [4]School of Engineering, Texas A&M, College Station, TX, USA. [5]Center for Neuroregeneration, Department of Neurosurgery, Houston Methodist Research Institute, Houston, TX, USA. [6]These authors contributed equally: Jesus G. Cruz-Garza, Lokeshwar S. Bhenderu. ✉e-mail: jgcruzgarza@houstonmethodist.org; lokeshwar.bhenderu@houstonmethodist.org; ahfaraji@houstonmethodist.org

barrier to achieving success in clinical applications. As an example, there is often increased interstitial pressure within brain tumors compared to the surrounding tissue and vasculature[26], which makes it difficult for the therapeutic agent to access the instertitum with CED alone[27].

Electrokinetic convection-enhanced delivery (ECED) is a delivery method based on the use of electromotive forces that attempts to address many of the challenges of conventional pressure-driven CED to enhance drug delivery. ECED permits bulk interstitial fluid flow without the need for a pressure gradient and allows for directed infusion of therapeutic agents along an electrical current path[24,28].

The use of electromotive forces to enhance the delivery of drugs has been used in other fields with FDA-approved applications for transdermal applications[29]. ECED in the CNS works by applying an external electric field between two electrodes[24]. One of the electrodes can be the infusion cannula itself. The electric field induces electroosmosis in the brain tissue, as a bulk fluid flow between the electrodes[28,30], due to the brain's non-zero zeta potential[31,32]. For charged particles, this electric field also induces electrophoresis, the movements of ions in a solution passing an electric current. In brain tissue, electroosmosis directs bulk fluid flow for charged or uncharged particles[31–34], which include most therapeutic agents currently applicable to the CNS (i.e., uncharged, neutral, and non-polar molecules). In addition, ECED provides the possibility of enhanced delivery of new classes of possible drugs (e.g., slightly negatively or positively charged molecules) through electrophoresis[24,34].

The use of ECED to deliver a solute to large tissue areas has been demonstrated in hydrogels[28], in hippocampal slice cultures[31,32], and as proof-of-concept in vivo[24]. In prior brain tissue experiments, the ECED setup was comprised of two silica capillaries inserted into the brain parenchyma[24]. These capillaries were connected to reservoirs of a HEPES (4-(2-hydroxyethyl)-1-piperazineethanesulfonic acid)-based saline solution (HBSS) that allowed the tip of the capillaries to act as electrodes with an electrical current between them.

Here, we expand on these studies by investigating the ability of ECED to enable intraparenchymal penetration from the surface of the brain using a doped biocompatible acrylic acid/acrylamide hydrogel[33]. We aim to provide proof of principle for the use of ECED as an infusion mechanism from a surface hydrogel to the brain using an external electric field, without the use of a pressure gradient as in CED.

We quantified the ECED-driven infusion of an uncharged fluorophore by measuring the infusion distance, linear infusion profile, fluorescence intensity, and area of infusion to show the feasibility of ECED to direct the infusion of a fluorophore into the rat brain in ex vivo and in vivo experiments, from a hydrogel reservoir at the surface of the brain, compared to diffusion alone. This method eliminates the need for an infusion cannula to deliver therapeutics by directing the infusion agent from a surface hydrogel toward the counter-cannula inside the brain. This method demonstrates the potential to provide rapid conveyance of therapeutics from doped hydrogels which can be placed at the cortical surface or along surgical resection cavities to allow for increased coverage of tumor margins.

## Results
### Experimental design for ECED-based infusion from hydrogel reservoir
The delivery of solutes to the brain through ECED requires an external electric field to deliver therapeutic agents through both electroosmosis and electrophoresis. In this experiment, we used the Texas Red 3 kDa dextran conjugate (TR3) fluorophore, a zwitterionic molecule, as the infusate. A poly(acrylamide-co-acrylic acid) hydrogel, with electrokinetic properties comparable to brain tissue[33], was doped with TR3 and placed at the surface of the brain. An electric field was established from a capillary connected to the surface of the hydrogel to a second capillary inside the brain, both filled with HBSS. The ECED-driven solute infusion was studied for a current of $I = 50\,\mu A$, and a time of 30 min[24,28]. Diffusion-only control trials were performed with the same setting, but without an electric field applied ($I = 0\,\mu A$). The flow of TR3 resulting from the

implementation of ECED to infuse the fluorophore from a hydrogel reservoir relied mostly on electroosmosis, with minor contribution from electrophoresis in this solute[34]. Ex vivo ($N = 18$) and in vivo ($N = 12$) trials were conducted on rodent brains over a 30-min intervention period. The experimental setup for the study is shown in Fig. 1.

Different exposure time images were used in the data analysis for in vivo (400 ms exposure) and ex vivo (200 ms exposure) studies based on their unique fluorescence intensity profiles due to tissue permeability differences in vitro and in vivo[35]. The data and statistical analysis for 200 ms and 400 ms exposure times for all trials is presented in Supplementary Table 1.

Linear plot profiles, perpendicular to the brain surface were used to determine depth of penetration from the surface of the brain. The depth of fluorophore penetration was determined as the distance from the surface of the brain where the fluorescence reached a certain threshold relative to the maximum intensity. We used 10% and 30% thresholds as an approximation of the concentration of the fluorophore that would be infused into the brain. These parameters were chosen to analyze the concentrated (30%) and diluted (10%) spread of the fluorophore.

From the linear plot profiles, we analyzed fluorescence intensity as a function of infusion distance in the brain, and the displacement of the fluorescence peak, measured as the distance from the surface of the brain where the maximum fluorophore intensity was observed. The fluorescence peak displacement indicates the bulk displacement of the infusion agent inside the brain. The linear infusion distance covered by the fluorophore at the 10% threshold of maximum intensity is abbreviated as $d_{10}$, and the linear infusion distance covered by the fluorophore at a threshold of 30% of maximum intensity is abbreviated as $d_{30}$. See Methods, Image data extraction, and inter-rater reliability for a visual representation of the linear profile outcome measurements.

The infusion area covered by the fluorophore from maximum intensity to 10% and 30% of the maximum was analyzed. The quantification of the infusion area allows for the concrete assessment of heterogeneous fluorophore distribution inside the brain. The area covered by the fluorophore at the 10% threshold of maximum intensity is abbreviated as $A_{10}$, and the area covered by the fluorophore at a threshold of 30% of maximum intensity is abbreviated as $A_{30}$.

### Hydrogel reservoir electroosmotic-driven infusion in vivo
The in vivo experimental trials were analyzed with the same metrics as the ex vivo counterparts. The focus of this analysis is to validate the directed infusion of the fluorophore from the hydrogel reservoir at the surface of the brain into the brain parenchyma. ECED trials showed significantly higher infusion distance and infusion area compared to diffusion-only control trials in the outcome metrics: distance to 10% of maximum intensity ($p < 0.05$), distance to 30% of maximum intensity ($p < 0.05$), fluorescence peak displacement ($p < 0.05$), area within 10% of maximum intensity ($p < 0.05$) (Table 1). The fluorescence peak intensity ($p < 0.40$) and area within 30% of the maximum intensity ($p = 0.16$) did not reach statistically significant difference between conditions. A clear effect of higher infusion distance was observed for ECED trials compared to diffusion-only control.

Figure 2a shows the mean and 95% CI for the linear intensity profiles taken from all the experimental trials from the surface of the brain at 0.00 mm to a chosen distance of 2.50 mm, as there was no fluorophore observed past this distance in the experimental trials. The ECED intervention showed statistically significant ($p < 0.05$) larger infusion fluorescence from 0.00 to approximately 2.00 mm and minimal differences up to 2.50 mm. These results are found strongly in the 0.40–1.5 mm range ($p < 0.01$). There was a higher concentration of infused fluorophore at these penetration distances, compared to the diffusion-only control.

The $d_{10}$ was significantly higher for in vivo ECED compared to the in vivo control condition, $t(10) = 3.14$, $p = 0.01$. The $d_{10}$ distance was $0.49 \pm 0.11$ mm larger in ECED trials than in control trials. The mean $d_{10}$ for ECED trials was $0.88 \pm 0.11$ mm from the surface of the brain, and it reached $0.39 \pm 0.11$ for the diffusion-only control (Fig. 2b).

## Ex vivo

**a** Experimental setup

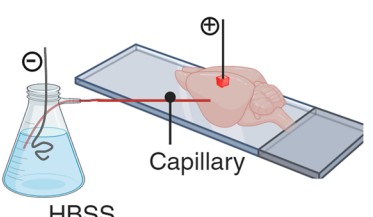

**b** ECED configuration

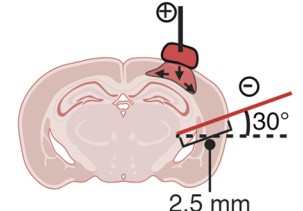

**c** Experimental conditions

T=30 min, I=0 µA.
**Control**

T=30 min, I=50 µA.
**ECED**

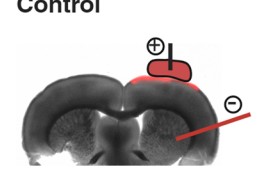
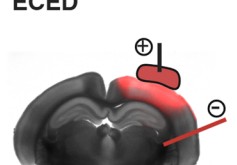

## In vivo

**d** Experimental setup

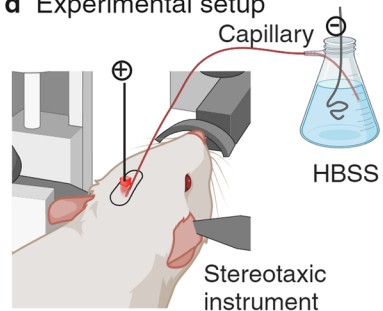

**e** ECED configuration

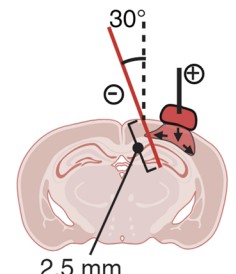

**f** Experimental conditions

T=30 min, I=0 µA.
**Control**

T=30 min, I=50 µA.
**ECED**

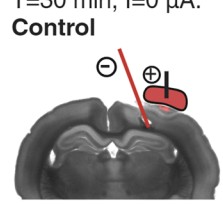
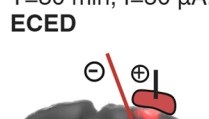

**Fig. 1 | Schematic of experimental setup for ECED transport. a** Ex vivo brain placed over a microscope slide under the stereoscope. **b** Configuration of hydrogel placement, capillary insertion, and schematic representation of the expected fluorophore infusion inside the brain. **c** Experimental conditions of diffusion-only control and ECED, with a representative example from each, created by merging a brightfield image with the fluorescent image. **d** In vivo subject placed on the stereotaxic device, showing a single craniotomy where the hydrogel and capillary were placed. **e** Detailed configuration of hydrogel and capillary placement in coronal view, with a schematic representation of the expected fluorophore infusion inside the brain. **f** Experimental conditions of diffusion-only control and ECED, with a representative example for each.

The distance $d_{30}$ was significantly higher in in vivo ECED compared to the in vivo control condition, $t(9) = 2.41$, $p < 0.05$. The $d_{30}$ distance was $0.23 \pm 0.18$ mm larger in ECED trials than in control trials. The mean $d_{30}$ for ECED trials was $0.33 \pm 0.07$ mm from the surface of the brain, and it reached $0.10 \pm 0.06$ for the diffusion-only control (Fig. 2b).

The area of infusion observed using the threshold of 10% of the maximum intensity ($A_{10}$) was significantly higher for in vivo ECED compared to the in vivo control condition, $t(10) = 2.84$, $p < 0.05$. The mean $A_{10}$ was $0.95 \pm 0.24$ mm² larger in ECED trials than in control trials. The mean $A_{10}$ for ECED trials was $1.30 \pm 0.26$ mm² from the surface of the brain, and it reached $0.35 \pm 0.21$ mm² for the diffusion-only control (Fig. 2d).

There was not a significant difference in the area of infusion observed using the threshold of 30% of the maximum intensity ($A_{30}$) fluorescence between in vivo ECED and in vivo control conditions, $t(9) = 1.52$, $p = 0.16$. The $A_{30}$ distance was $0.16 \pm 0.07$ mm² larger in ECED trials than in control trials. The mean $A_{30}$ for ECED trials was $0.22 \pm 0.09$ mm² from the surface of the brain, and it reached $0.07 \pm 0.06$ mm² for the diffusion-only control (Fig. 2d).

There was a significant difference in fluorescence peak displacement between in vivo ECED and in vivo control conditions $t(8) = 3.32$, $p = 0.01$. The peak displacement was $0.13 \pm 0.03$ mm higher for ECED than control. It reached $0.21 \pm 0.03$ mm for ECED and $0.08 \pm 0.02$ mm for the control condition. (Fig. 2c).

There was not a significant difference in fluorescence peak intensity between in vivo ECED and in vivo control conditions, $t(10) = 0.87$, $p = 0.40$. The mean fluorescence peak intensity was $16.32\% \pm 13.17\%$ total percentage points larger in in vivo ECED trials than in in vivo control trials. The mean fluorescence peak intensity for ECED trials was $51.45\% \pm 12.39\%$, and it reached $35.13\% \pm 13.95\%$ for the diffusion-only control (Fig. 2e).

### Hydrogel reservoir electroosmotic-driven infusion ex vivo

The infusion in ECED vs diffusion-only control in ex vivo trials did not reach statistical significant differences for the metrics: distance to 10% ($p = 0.21$) and 30% ($p = 0.07$) of the maximum intensity, area covering up to 10% ($p = 0.32$) and 30% ($p = 0.12$) of the maximum intensity, fluorescence peak displacement ($p = 0.54$). The fluorescence peak intensity ($p < 0.05$) reached statistical significance in this comparison (Table 2). A trend for higher infusion in ECED trials compared to control was consistently present.

Figure 3a shows the mean and 95% CI for the linear intensity profiles taken from all the experimental trials from the surface of the brain at 0.00 mm to a standard distance chosen displayed of 2.50 mm. The ECED intervention showed larger infusion fluorescence from 0.00 to approximately 1.50 mm and minimal differences up to 2.50 mm. The unequal variances t-test indicated that there is a significant difference in fluorescence intensity, an indication of infusion volume, between the control vs ECED distributions at a threshold of $p < 0.05$ before in the range between 0.00 and 1.50 mm distance. There was a strong significant difference ($p < 0.01$) in fluorescence intensity between 0.1 and 0.5 mm of infusion distance between the experimental conditions, which reinforces the clear trend for higher fluorescence intensity in ECED trials observed. The distance to 10% of the maximum intensity ($d_{10}$) was higher for ex vivo ECED compared to the ex vivo control condition, $t(16) = 1.31$, $p < 0.21$. The mean $d_{10}$ was $0.16 \pm 0.09$ mm larger in ECED trials than in ex vivo control trials. The mean $d_{10}$ for ECED trials was $0.71 \pm 0.09$ mm from the surface of the brain, and it reached $0.55 \pm 0.09$ mm for the diffusion-only control (Fig. 3b).

The distance to 30% of the maximum intensity ($d_{30}$) was likewise higher in ex vivo ECED compared to the ex vivo control condition, $t(15) = 1.96$, $p = 0.07$. The $d_{30}$ distance was $0.20 \pm 0.07$ mm larger in ECED

**Table 1 | In vivo statistics for comparison of means between ECED and control conditions**

| Experiment | Exposure time (ms) | Outcome measurement | Units | Control | ECED | ECED - Control | *p*-value |
|---|---|---|---|---|---|---|---|
| In vivo | 400 | Distance 10% of max. intensity | mm | 0.39 ± 0.11 | 0.88 ± 0.11 | 0.49 ± 0.11 | 0.01[a] |
| In vivo | 400 | Distance 30% of max. intensity | mm | 0.10 ± 0.06 | 0.33 ± 0.07 | 0.23 ± 0.06 | 0.04[a] |
| In vivo | 400 | Fluo. peak displacement | mm | 0.08 ± 0.02 | 0.21 ± 0.03 | 0.13 ± 0.02 | 0.01[a] |
| In vivo | 400 | Fluo. peak intensity | % | 35.13 ± 13.95 | 51.45 ± 12.39 | 16.32 ± 13.17 | 0.40 |
| In vivo | 400 | Area within 10% of max. intensity | mm² | 0.35 ± 0.21 | 1.30 ± 0.26 | 0.95 ± 0.23 | 0.02[a] |
| In vivo | 400 | Area within 30% of max. intensity | mm² | 0.07 ± 0.06 | 0.22 ± 0.09 | 0.16 ± 0.07 | 0.16 |

[a]indicates an unequal variances *t*-test *p*-value of $p < 0.05$. Data presented as mean ± standard error of the mean.
Statistical test to calculate *p*-value: Welch's *t*-test for unequal variances.

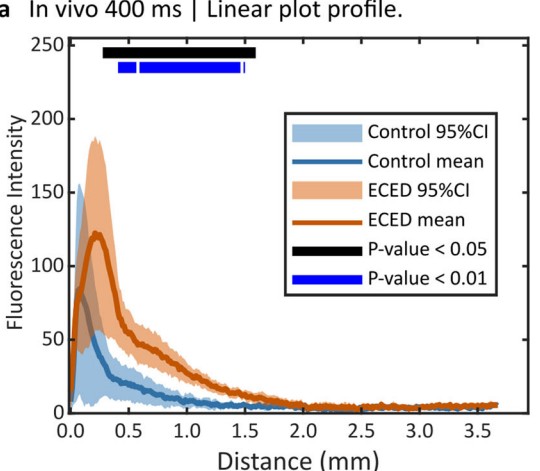

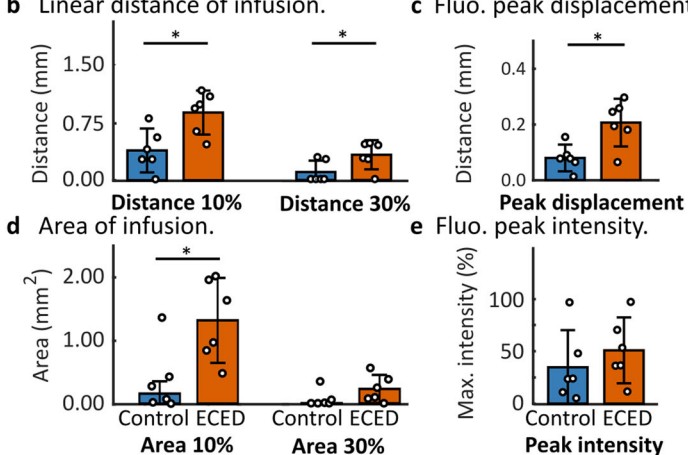

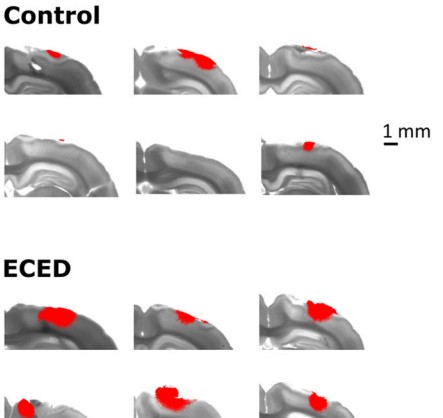

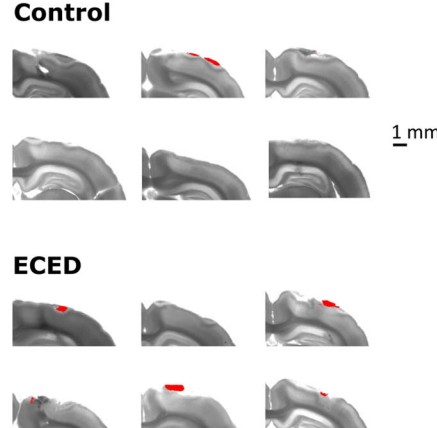

**f** Fluorescence. 10% of maximum intensity or greater.

**g** Fluorescence. 30% of maximum intensity or greater.

**Fig. 2 | In vivo results at 400 ms exposure time, $N = 12$ biologically independent subjects. a** Mean and 95% confidence intervals (CI) of the plot profiles that measure the fluorescence intensity of the fluorophore infusion from the surface of the brain, identified in the plots as distance = 0.00 mm, perpendicularly towards inside the brain. **b** Distribution of fluorophore infusion distance, measured from the surface of the brain to the point where the fluorescence intensity reached 10% of the maximum intensity ($d_{10}$) and 30% of the maximum intensity ($d_{30}$). **c** Displacement from the surface of the brain, for the highest value of fluorescence intensity in each trial. **d** Area covered by fluorophore infusion, at a threshold of 10% of the maximum

intensity ($A_{10}$) and 30% of the maximum intensity ($A_{30}$). **e** Fluorescence peak intensity value in each trial, as a percentage of the highest peak in all trials. The bar graph indicates the mean, and the error bars correspond to the 95% CI.
**e** Fluorescence peak intensity value in each trial, as a percentage of the highest peak in all trials. **f** Fluorescence images overlaid upon brightfield images for the brain slice selected for analysis, with the intensity threshold set at 10%, and **g** the intensity threshold set at 30% of maximum intensity. Significant differences are indicated by ** for $p < 0.01$, and * for $p < 0.05$.

**Table 2 | Ex vivo statistics for comparison of means between ECED and control conditions**

| Experiment | Exposure time (ms) | Outcome measurement | Units | Control | ECED | ECED - Control | *p*-value |
|---|---|---|---|---|---|---|---|
| Ex vivo | 200 | Distance 10% of max. intensity | mm | 0.55 ± 0.09 | 0.71 ± 0.09 | 0.16 ± 0.09 | 0.21 |
| Ex vivo | 200 | Distance 30% of max. intensity | mm | 0.39 ± 0.06 | 0.60 ± 0.08 | 0.20 ± 0.07 | 0.07 |
| Ex vivo | 200 | Fluo. peak displacement | mm | 0.17 ± 0.03 | 0.19 ± 0.03 | 0.03 ± 0.03 | 0.53 |
| Ex vivo | 200 | Fluo. peak intensity | % | 47.00 ± 7.43 | 73.72 ± 7.04 | 26.73 ± 7.21 | 0.02[a] |
| Ex vivo | 200 | Area within 10% of max. intensity | mm$^2$ | 1.29 ± 0.42 | 1.87 ± 0.38 | 0.57 ± 0.40 | 0.32 |
| Ex vivo | 200 | Area within 30% of max. intensity | mm$^2$ | 0.11 ± 0.07 | 0.44 ± 0.18 | 0.33 ± 0.13 | 0.12 |

[a]indicates an unequal variances *t*-test *p*-value of $p < 0.5$. Data presented as mean ± standard error of the mean.
Statistical test to calculate *p*-value: Welch's *t*-test for unequal variances.

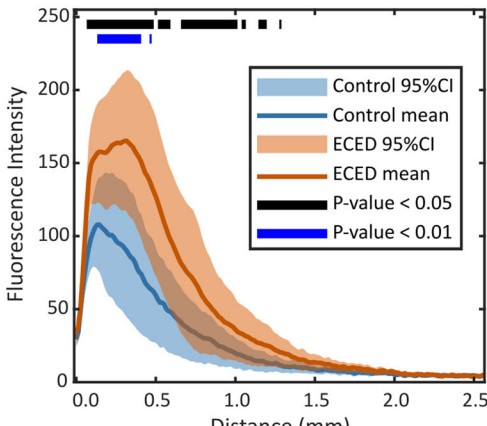

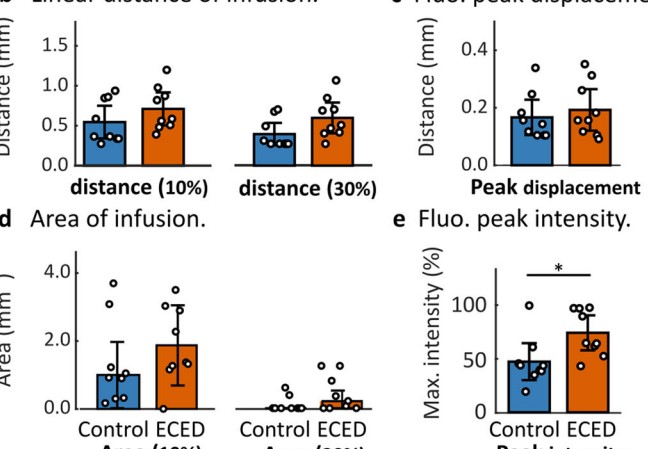

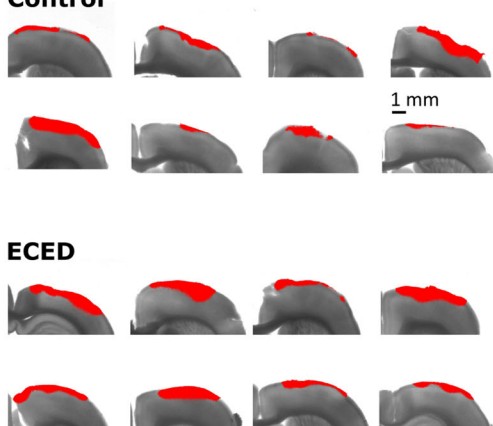

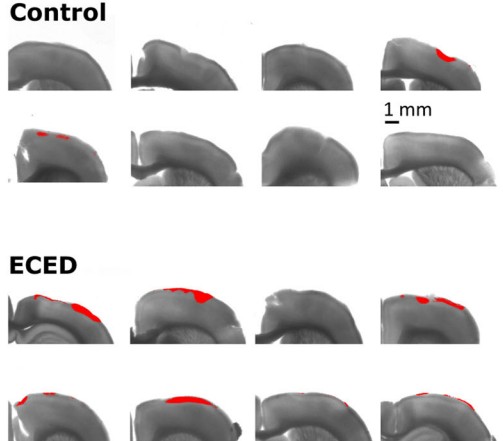

**Fig. 3 | Ex vivo results at 200 ms exposure time, *N* = 18 biologically independent subjects. a** Mean and 95% confidence intervals (CI) of the plot profiles that measure the fluorescence intensity of the fluorophore infusion from the surface of the brain, identified in the plots as distance = 0.00 mm, perpendicularly towards inside the brain. **b** Distribution of fluorophore infusion distance, measured from the surface of the brain to the point where the fluorescence intensity reached 10% of the maximum intensity (d$_{10}$) and 30% of the maximum intensity (d$_{30}$). **c** Displacement from the surface of the brain, for the highest value of fluorescence intensity in each trial.

**d** Area covered by fluorophore infusion, at a threshold of 10% of the maximum intensity (A$_{10}$) and 30% of the maximum intensity (A$_{30}$). **e** Fluorescence peak intensity value in each trial, as a percentage of the highest peak in all trials. The bar graph indicates the mean, and the error bars correspond to the 95% CI.
**f** Fluorescence images overlaid upon brightfield images for the brain slice selected for analysis, with the intensity threshold set at 10%, and **g** the intensity threshold set at 30% of maximum intensity. Significant differences are indicated by ** for $p < 0.01$, and * for $p < 0.05$.

trials than in ex vivo control trials. The mean $d_{30}$ for ECED trials was $0.60 \pm 0.08$ mm from the surface of the brain, and it reached $0.39 \pm 0.06$ mm for the diffusion-only control (Fig. 3b).

The area of infusion observed using the threshold of 10% of the maximum intensity ($A_{10}$) was higher for ex vivo ECED compared to the ex vivo control condition but did not reach statistical significance, $t(16) = 1.02$, $p = 0.32$. The mean $A_{10}$ was $0.57 \pm 0.40$ mm² larger in ECED trials than in control trials. The mean $A_{10}$ for ECED trials was $1.87 \pm 0.38$ mm from the surface of the brain, and it reached $1.29 \pm 0.42$ mm for the diffusion-only control (Fig. 3d).

The area of infusion observed using the threshold of 30% of the maximum intensity ($A_{30}$) was higher in ex vivo ECED compared to the ex vivo control condition, $t(11) = 1.71$, $p = 0.12$. The $A_{30}$ was $0.33 \pm 0.12$ mm² larger in ECED trials than in control trials. The mean $A_{30}$ for ECED trials was $0.44 \pm 0.18$ mm from the surface of the brain, and it reached $0.11 \pm 0.07$ for the diffusion-only control (Fig. 3d).

There was not a significant difference in fluorescence peak displacement between ex vivo ECED and ex vivo control conditions $t(16) = 0.03$, $p = 0.54$. The fluorescence peak displacement from the surface of the brain reached $0.19 \pm 0.03$ mm for ECED, and $0.17 \pm 0.03$ mm for the control condition (Fig. 3c).

There was a significant difference in fluorescence peak intensity between ex vivo ECED and ex vivo control conditions $t(16) = 2.61$, $p < 0.05$. The mean fluorescence peak intensity was $26.73\% \pm 7.23\%$ total percentage points larger in ex vivo ECED trials than in ex vivo control trials. The mean fluorescence peak intensity for ECED trials was $73.72\% \pm 7.04\%$, and it reached $47.00\% \pm 7.43\%$ for the diffusion-only control (Fig. 3d).

## Discussion

ECED utilizes an external electric field to deliver therapeutic agents to the brain through electroosmosis and/or electrophoresis. Our current study evaluates the capacity of electroosmosis to convey a small (3 kDa), neutral, fluorescent dextran conjugate from a doped hydrogel and across the cortical surface to result in intraparenchymal delivery. Electroosmotic flow to deliver molecules to the brain has been previously shown with positively charged small molecules[24]. These results show that the electroosmotic vector is enough to convey not only positively charged molecules but neutral-charged molecules into the brain parenchyma.

In our study, however, there is a small contribution to the ECED infusion that is due to electrophoretic mobility. The dextran fluorophore used in the current study, TR3, has been shown to have a small positive electrophoretic mobility of $[2.56 \pm 0.04]*10^{-9}$ m²/(V s), as determined by a Taylor dispersion method. Its transport by an electric field is influenced 7.6 times greater by electroosmosis than electrophoresis in these hydrogels with $-24.7$ mV zeta potential[34].

The ex vivo and in vivo ECED trials implemented to deliver small fluorescent molecules to the brain showed a consistently significant greater fluorescence intensity along the plot profiles compared to their respective diffusion-only control conditions, indicating consistent penetration into brain tissue.

The peak displacement of the fluorescence intensity is another important measurement, as it indicates the bulk displacement of the infusion agent inside the brain. The peak displacement was interpreted as the distance from the surface of the brain where the highest concentration of fluorophore was observed. The difference in peak displacement between the ECED and control trials only reached significance for the in vivo experiment. The greater peak displacement distance in the in vivo experiment indicates that a greater amount of fluorophore was moved deeper into the parenchyma with ECED than control.

In vivo trials of ECED showed consistent higher concentration of infusion at 0.3–1.5 mm of linear penetration in the brain, compared to controls ($p < 0.05$), with most of this range showing strongly significant differences ($p < 0.01$) (Fig. 2a). Other metrics consistently showed significant differences between conditions (Fig. 2b–g). Ex vivo trials showed strongly significant differences between conditions between 0.1 mm and

0.5 mm ($p < 0.01$), and significant differences in infusion profiles from 0.1 to 1.5 mm ($p < 0.05$) (Fig. 3a). Although ex vivo ECED trials consistently showed higher infusion penetration, other metrics such as area of infusion to 10% and 30% of maximum intensity, and distance of infusion to 10% and 30% of maximum intensity did not show significant differences (Fig. 3b–g). The effect of the intervention was stronger in in vivo trials than in ex vivo trials.

The area of infusion was larger in ECED compared to control conditions for in vivo and ex vivo trials consistently, but it was only statistically significant for $A_{10}$ in in vivo trials. ECED facilitated further penetration of the fluorophore in the brain tissue, which was especially evident in in vivo trials. For ex vivo trials, we observed a large transversal spread of the fluorophore along the surface of the hydrogel that lay on the brain in both experimental conditions. From there, although the linear intensity profiles (Fig. 3a) showed higher intensity of the fluorophore that penetrated further into the brain, the area covered by this fluorophore at the thresholds selected was not statistically significantly larger. For in vivo trials, both the fluorophore concentration at deeper penetration distances (Fig. 2a) and the area covered (Fig. 2d) were statistically significantly larger in ECED compared to control.

Potential reasons for these differences include changes in tissue porosity and structure in the in vivo and ex vivo brains[36] and unreplenished interstitial fluid as the ex vivo brain were isolated without cerebrospinal fluid. Finally, for the ex vivo trials, multiple unfixed brains were harvested at similar times and stored at 4 °C, but trials were run sequentially. The average time from sacrifice to the start of the trial was $3.28 \pm 0.30$ h, contributing to differences in tissue permeability between trials. This was not an issue with the in vivo experiments as each subject was sacrificed immediately after the experiment intervention, and imaged.

Extracellular space in normative conditions has shown that the ratio of extracellular space in the brain is $\alpha = 0.20$[36,37], but when brain slices are placed in hypoxic media for 10–30 min, $\alpha$ decreases to $\alpha = 0.15$ in 5–10 min and $\alpha = 0.12$ in 10–20 min[36]. The effects of tissue porosity likely resulted in a lower brain permeability in ex vivo trials compared to in vivo. In addition, taking in vitro analyses as an example, although there is a correlation between drug and solute permeability in studies in vitro and in vivo[38], the permeability coefficients are 150-fold higher in vitro than for in vivo models of the brain[35].

Volumetric analyses could not be accurately determined due to the 500 μm thickness of the brain slices. This large thickness led to a loss of axial spatial resolution in the brain. Future studies will evaluate the volumetric ECED infusion profile via real-time magnetic resonance imaging[25].

Previous experiments showing ECED in vivo were completed with an infusion cannula that was inserted into the brain. The infusion cannula acted as the positive electrode while a counter-cannula, also inserted into the brain parenchyma, acted as the negative electrode. A current was induced between the tips of the two cannulas and molecules were shown to move up to 4 mm using ECED when using a 25% threshold value of the maximum fluorescent intensity[24]. Our current experiment explores a therapeutic delivery method based on an electric field gradient, with a hydrogel placed over the brain surface and with the use of only one cannula inside the brain; with no intraparenchymal infusion cannula. The positive electrode sits inside a doped hydrogel on the surface of the brain and ECED allows for infusion into the brain. Therefore, there is a significant barrier for the ECED to overcome at the surface of the brain. We used a hydrogel with a similar zeta potential as the brain to decrease the potential barrier between the gel and the brain[34].

Like CED, this method will allow locally high concentrations of the drug to be delivered to specific portions of the brain while reducing complications from systemic toxicity[39]. Importantly, we introduce a potential treatment method by not using an infusion cannula and instead using a doped hydrogel to store and deliver therapeutic agents.

There have been previous studies using hyaluronan/methyl cellulose hydrogels for the delivery of erythropoietin after stroke injury[40,41]. These studies demonstrate the effectiveness of an epicortical hydrogel to deliver a

therapeutic agent via diffusion alone. ECED may improve drug delivery in such epicortical hydrogels if these hydrogels can be engineered to support electroosmosis.

ECED also has applications in neuro-oncologic surgery. Clinical studies have shown the effectiveness of diffusion-based polymer depots containing chemotherapeutic drugs placed inside tumor resection cavities at the time of surgery[42,43]. ECED may be able to augment the penetration of such drugs and enhance delivery to surrounding parenchyma. Instead of using previously described absorbable wafers, clinicians may use biocompatible doped hydrogels at the time of surgery to store local chemotherapeutics. Future studies will need to investigate whether the polymer substances previously described are compatible with ECED. Although polymers are porous materials, the zeta-potentials are different from that of the brain, and this may be significant enough to alter the electrodynamics. The effectiveness of ECED could be assessed histologically, by examining the tissue penetration of neuro-oncologic drugs, or anti-inflammatory agents around the resection cavity, compared to current methods of diffusion.

In future work, we will explore the degree to which therapeutic drugs can also be delivered into the brain without the current limitations for infusion cannulas such as tissue damage, backflow, and limited duration due to clogging[23,44,45].

Previous experiments demonstrating ECED in hydrogel and hippocampal slice cultures[28,34], and proof-of-principle application in vivo[24], have utilized an infusion capillary to deliver macromolecules into brain tissue. The bulk flow resulting from electroosmosis and electrophoresis in previous experiments show highest velocities near the infusion capillary and the counter electrode capillary[28,34]. The electroosmotic velocity is proportional to the electric field magnitude[31], and therefore it is proportional to $1/z^2$ from the capillary tip, modeled as a point source; where z is the distance from the point source. In our experiment, a hydrogel of $9 \times 10^{-6}$ m$^2$ surface area in contact with the brain was used with a counter electrode being a 100 μm ID capillary ($7.85 \times 10^{-9}$ m$^2$ surface area), the electric field magnitude can be approximated by Equation (1), with r: radius of circle of surface area in contact with the brain, σ: surface charge density, z: distance to point in its central axis, $\epsilon_0$: vacuum permittivity. Our hydrogel setup was a square (3 mm × 3 mm) in contact with the brain, but we used the shape of a charged disk in for simplicity.

$$E = \frac{\sigma}{2\epsilon_0} \left( 1 - \frac{z}{\sqrt{z^2 + r^2}} \right) \qquad (1)$$

The electric field between a charged layer at the surface of the brain and the counter electrode inside the brain would follow approximately Eq.(1) in the adjacency of the hydrogel placement. In our experiment, neutral-charged macromolecules were infused through electroosmosis into the brain parenchyma, with the electric field strength decreasing at a rate proportional to the second term of Eq.(1). The rate of electric field decay in our setup is slower than the $1/z^2$ rate from a point source (cannula tip). Therefore, the infusion of macromolecules from a hydrogel reservoir is expected to maintain higher electroosmotic velocity for longer distances.

The hydrogel-based ECED technique also offers lower electrical resistance compared to the two cannula tips setup. The voltage used was between the magnitudes of 925–1300 V for ex vivo and 1000–1200 V for in vivo trials. The hydrogel provides an attractive method for drug delivery with the use of an electric field that can reach voltage ranges closer to voltage sources available in the operating room. Future work will explore counter-cannula placement, cannula diameter, and electric current magnitudes that optimize voltage ranges and macromolecule delivery to the brain.

The motivation for this study was to provide an alternative method for solute delivery to the brain, through electroosmosis, that does not rely on a pressure gradient to guide the delivery of the therapeutic agent. This report provides evidence for the effectiveness of ECED to deliver an uncharged macromolecule (3 kDa) to the brain parenchyma from a hydrogel reservoir at the surface of the brain. Infusion from a hydrogel reservoir would not be possible with pressure-driven CED.

We envision this ECED configuration as a candidate therapeutic agent delivery mechanism for the treatment of glioblastoma. Pathophysiological theory of glioma ontogeny suggests that initiation points are distributed throughout the subcortical white matter[46], with many glioblastoma tumors appearing in contact with the inner-most cortical layer[47]. The use of hydrogel systems to deliver various types of therapeutics, ranging from polymeric micelles to viral vectors, has been studied for glioblastomas as well as other types of brain tumors[48,49]. The addition of ECED would allow for enhanced delivery of these therapeutic agents, possibly even at the time of neurosurgical resection, and therefore increase the efficacy of these treatments. For example, an in situ hydrogel can be placed in a tumor resection cavity to allow for the therapeutic delivery of molecules. The use of doped hydrogels in resection cavities has already appeared in literature[48], but the use of ECED can potentially increase the efficacy of these treatment options. Further applications of ECED can be used to decrease neuroinflammation after traumatic brain injuries by delivering anti-inflammatory agents, such as dexamethasone, to areas of injury using a doped hydrogel and ECED[50]. Overall, our experiments show that ECED can potentially be used to augment treatments which use doped hydrogels by increasing the amount of drug delivery to the region of interest. Furthermore, the principles behind ECED are also being used to manipulate fluid flow in the brain and reduce the harmful effects of cerebral edema[51,52]. Our current findings can also be compounded with these studies to better control the flow of fluids in the brain and gain, ultimately, control of drug distribution in the brain.

In this study, the counter cannula was placed below the hydrogel, inside the brain, at a fixed angle and distance of insertion. We evaluated the perpendicular infusion of the fluorophore towards the counter cannula. Further research is needed to assess the directionality of delivery to infuse the agent along a preferred trajectory inside the brain from a hydrogel reservoir ECED configuration: e.g. for transversal and oblique infusion directions. In addition, the expected volume of distribution and tumor coverage can be optimized by the systematic analysis of electric current intensity, time of intervention, infusate concentration, and capillary size.

## Materials and methods
### Study design
**Sample size**. The sample sizes for this study were estimated from preliminary data of ECED in hydrogel and adjusted after the initial set of experiments for ECED in brain observations. The expected effect size was estimated as $d = 2.04$ based on previous experiments[28] demonstrating ECED in hydrogel with similar electroosmotic properties as the brain. This effect size is estimated for the comparison of ECED at 25 μA and the diffusion-only control condition, 0 μA. For error probability alpha = 0.05, and Power (1 - β) = 0.80, the recommended sample size is at least $N = 5$ per group. Given that this preliminary study of effect size was done in hydrogel, we re-assessed the number of subjects per group with intermediate data of three subjects per group for ex vivo and in vivo experiments. The estimated effect size for ex vivo trials was $d = 1.6$, yielding an updated recommended sample size of $N = 8$ per group. The estimated effect size for in vivo trials was $d = 1.8$, resulting in a recommended sample size of $N = 6$ per group. Data collection was stopped when the specified number of subjects was obtained.

**Data inclusion/exclusion criteria**. Experiments that were carried out to completion with no significant damage to the brain from the craniotomy procedure were included in the analysis: i.e. 30 min of the intervention with ECED or diffusion-only control at the specified experimental electrical current conditions. Experiments where the current fluctuated, due to the presence of air bubbles or other anomalies in the capillaries, were included in the analyzed results if these fluctuations were lower than the desired current (50 μA for ECED) and less than three minutes of the total intervention time. The three-minute cutoff was determined post-hoc. Thirteen out of fourteen trials of in vivo trials met the inclusion criteria, with one included trial presenting current fluctuation under 1 min in duration. For ex vivo trials, all eighteen samples met this inclusion

criteria, with one included ECED trial presenting fluctuating current for 3 min of the total intervention duration.

**Outliers.** Experimental trial outliers were defined as those brain slice images that did not fit the expected fluorophore distribution, potentially from contamination. One in vivo ECED trial showed brain slice fluorescence over most of the brain making proper analysis not possible. The oversaturation of fluorescence was potentially due to paraformaldehyde or fluorophore contamination. This trial was excluded from the analysis, yielding a total of $N = 12$ subjects for analysis in this experiment.

In each statistical comparison, outliers were defined as those data points that fell beyond three standard deviations of the mean. No outliers were observed in the data points obtained for statistical analysis.

**Selection of endpoints.** The endpoint of the experimental intervention was the completion of the intervention time: 30 min.

**Research objectives.** The primary objective of this research was to measure fluorophore penetration inside the brain, comparing ECED-directed infusion with diffusion-only control. This experiment was implemented at a clinically relevant intervention time of 30 min. The electrical current selected for investigation was $I = 50\,\mu A$, a current that was estimated to provide verifiable results for ECED infusion[24], while minimizing the number of research subjects used.

In this experiment, we hypothesized that ECED ($I = 50\,\mu A$) would enhance macro-particle delivery into the brain parenchyma from a hydrogel reservoir of an infusion agent placed at the surface of the brain. This effect would be compared to a diffusion-only control with the same experimental setup, but current at $I = 0\,\mu A$. We determined three main outcome metrics to evaluate the infusion of the fluorophore in each condition: (1) distance of a fluorescence-threshold intensity to 10% and 30% of maximum intensity inside the brain, (2) fluorescence peak displacement inside the brain parenchyma, and (3) fluorescence intensity. The first two metrics would be measured with a linear intensity profile perpendicular to the brain surface.

The 10% and 30% thresholds were chosen as two arbitrary points to evaluate the effective concentration of the fluorophore as it dilutes in the brain. A threshold of 25% of the maximum intensity has been used in previous experiments[24], which is suitable for agents with a therapeutic index greater than 4. Supplementary Data 1 and Supplementary Table 1 contain the values for 50% of the maximum intensity as an additional metric point.

After the initiation of data analysis, we characterized the fluorescence intensity with respect to the linear profile distance inside the brain, which provided an overall visualization of the infusion profiles in each condition.

**Research subjects.** Sprague–Dawley rat cadavers, 7–9 weeks of age were used in the ex vivo trials. These subjects were donated from a different experiment, 15–20 min after they were sacrificed. These subjects were control subjects with no prior intervention. Exact age, weight, and sex were not available for these rats cadavers. All rats were euthanized via isoflurane and subsequently underwent a pancreatectomy. Carcasses were then placed into a plastic bag and placed at 4 °C.

Sprague–Dawley rats, 8–10 weeks of age, 6 males and 6 females, were used for the in vivo trials.

**Experimental design.** This was a randomized controlled laboratory experiment on rat subjects. The experiments were performed sequentially. There are two experiments reported: ECED in ex vivo subjects, and ECED in in vivo subjects. For each experiment, the study groups were the intervention group, ECED ($I = 50\,\mu A$), and the diffusion-only control group ($I = 0\,\mu A$). The experimental setup was the same between the ECED and control conditions. The brain was sliced at 500 μm thickness using a vibratome. The brain slices were observed and analyzed using a fluorescent microscope at 200 ms for ex vivo, and 400 ms for in vivo due to differences in fluorescence intensity. The slices with the highest

fluorescence intensity were selected for further analysis independently by two annotators. Perpendicular linear profiles were manually done by two independent annotators; and agreement between them was assessed with high inter-rater reliability scores. No further pre-processing was performed for the fluorescent images. The outcome metrics were estimated from these images.

**Randomization.** The trial order of experimental conditions (ECED or control) was randomized and defined before the start of the experiment. Experiments were conducted following this pre-determined randomization.

**Blinding.** The animal caretakers were blinded to the allocation sequence, before and after the intervention. The experimenters were not blinded during the experiment interventions, as they had to set the current at the corresponding level for ECED or control conditions. The subject IDs after the completion of experiments were blinded to the experimental condition they were assigned to. The order of experimental conditions and subject IDs were kept in a separate data file for matching purposes at the end of data extraction. The experimenters who assessed, measured, and quantified the results were blinded to the experimental condition of the data samples. The two annotators who selected the images for analysis and created the linear intensity profiles were blinded to the experimental conditions each trial belonged to. The annotators were blinded from each other.

## Solution preparations

Solution preparation followed the guidelines in previous research studying the electroosmotic properties of 25% acrylic acid-based hydrogels[33]. The following materials were purchased from Sigma–Aldrich: HBSS containing (mM) 143.4 NaCl, 5 HEPES, 5.4 KCl, 1.2 $MgSO_4$, 1.2 $NaH_2PO_4$, and 2.0 $CaCl_2$ was prepared with 18 MΩ purified water from a Milli-Q Advantage A10 system (Millipore, Billerica, MA), filtered, adjusted to pH 7.40, and stored at room temperature. The fluorophore used in this experiment was Texas Red dextran, 3 kDa (TR3). TR3 was dissolved in HBSS to make a 0.40 mM solution.

## Synthesis of hydrogel

A poly (acrylamide-co-acrylic acid) hydrogel, 25% acrylic acid, that has similar electrokinetic properties to brain tissue was used in our experiments, with a target zeta potential of $-24.7\,mV$[33]. The gel was placed in a covered Pyrex dish filled with approximately 25 mL HBSS solution and shaken lightly at room temperature overnight. The buffer solution was exchanged the following day with 25 mL of fresh HBSS and placed back in the shaker at room temperature for another 24 h. Finally, the HBSS was exchanged once more (25 mL), and the gel was placed in the 4 °C refrigerator where it completed the equilibration.

## Electrokinetic infusions ex vivo

Eighteen Sprague–Dawley rat cadavers, 7–8 weeks of age, with male and female specimens evenly distributed across conditions, were donated as ex vivo subjects for this experiment. The brain was extracted 3.28 ± 0.30 h of the sacrifice.

The brain was placed on a glass slide under the operative stereoscope (Fig. 1a). A $3 \times 3 \times 3$ mm TR3 doped hydrogel was placed on the left side of the brain. Using a stereotaxic instrument (Stoelting Co, Illinois, USA), a catheter was inserted laterally from the left side at 2.5 mm depth, at an angle of 30° from the table (Fig. 1b). The catheter was made from an 18 cm-long, 100 μm inside diameter, 360 μm outside diameter polyimide-coated fused silica capillary.

For ECED trials, a current of 50 μA was applied for a span of 30 min. In control trials, the same experimental setup was implemented, with a current of 0 μA (Fig. 1c). At the end of 30 min, the hydrogel piece was removed first, followed by removing the capillary. The brain was then dipped in a 4% paraformaldehyde solution for 5 min.

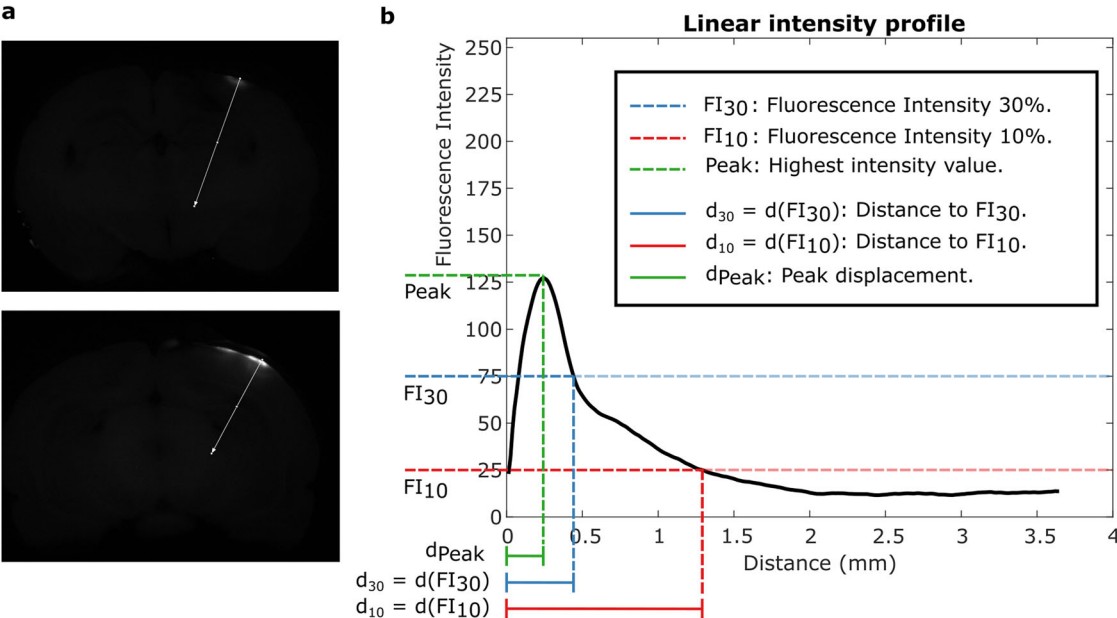

**Fig. 4 | Schematic visualization of the plot profile evaluation metrics for ECED-based infusion of molecules into the brain. a** Two separate examples of linear trajectories selected for evaluation. **b** Smoothed plot profile of one experimental trial, with the outcome variables of interest: Peak displacement $d_{Peak}$, distance to 30% of the maximum intensity $d_{30}$, and distance to 10% of the maximum intensity $d_{10}$.

## Electrokinetic infusions in vivo

The Institutional Animal Care and Use Committee (IACUC) at Houston Methodist Research Institute Institutional Animal Care and Use Committee approved the following procedures on protocol IS00006413. We have complied with all relevant ethical regulations for animal use.

Fourteen Sprague–Dawley rats with weights ranging from 189 to 385 g (average 266 g), age of 8–10 weeks, with males and females evenly distributed across conditions, were anesthetized with 2% isoflurane and reduced to a range of 0.5–1.5% for maintenance during the duration for the procedures. The rats were secured in a stereotaxic instrument. The rats were placed over a 37 °C homeothermic blanket.

A midline incision was made on the scalp to expose the bregma and lambda. Once the tissue was cut through, the tissue was retracted, and the blade was used to gently scrape any remaining periosteal and galeal membranes from the outside of the skull. At this point, the bregma could be clearly identified on the skull and the retractor was adjusted to reveal the lambda. Using a stereotaxic instrument, rats underwent a single craniotomy surgery on the left side of the skull that allowed for the positioning of a catheter at a 2.5 mm depth from the brain surface, at 30° insertion angle from the vertical line (Fig. 1d, e). Catheters were made from 18 cm-long, 100 μm inside diameter, 360 μm outside diameter polyimide-coated fused silica capillaries. A $3 \times 3 \times 3$ cm piece of hydrogel doped with TR3 was placed on top of the brain surface on the left side of the capillary at 1–2 mm distance from the capillary (Fig. 1e). A counter silver electrode was placed inside the hydrogel.

For ECED trials, a current of 50 μA was applied for a span of 30 min. In control trials, the same experimental setup was implemented, with a current of 0 μA. At the end of 30 min, the hydrogel piece was removed first, followed by removing the capillary (Fig. 1f).

The rats were perfused transcardially with 250 mL of 4% paraformaldehyde in phosphate-buffered saline solution for 5 min. The brain was then quickly extracted and cut at the cerebellum. Using a vibratome, the remaining part of the brain was sectioned into 500 μm thick slices. Two to three slices were taken for imaging while the slicing continued. Images were taken using fluorescence stereoscope. The time between infusion completion and imaging was approximately 20 min.

## Brain slicing in vibratome

An NVSLM1 vibroslice was used to slice the freshly harvested brain into 500 μm coronal cross-sections for imaging. To get the proper orientation, the brainstem and cerebellum were cut off, and the brain was glued using cyanoacrylate glue to the vibratome's podium on the freshly cut, flat side so that the frontal cortex was at the topmost point of the brain. The glue was allowed to dry and then a Phosphate-buffered saline (PBS) solution was added to the vibratome box until the entire brain was covered with PBS. A single-edge, 0.07 mm thick vibratory slicer blade was used to slice the brain. The vibratome is usually a quick setup that involves gluing the tissue to the platform and no encapsulation is needed for this technique.

## Brain imaging

A LEICA M165 FC stereoscope was used to take images of the coronal brain slices, Camera: LEICA DFC3000 G, Software: Leica Application Suite X, Leica Microsystems CMS GmbH. Brightfield (BF) and DSRed (excitation band: [525–560] nm, emission band: [590–650] nm) filter were utilized to visualize the brain. One image of a brain slice was taken with exposure under BF set at 40 ms. Four images were taken for fluorescence under different exposures: 100, 200, 400, and 800 ms. The images were collected and saved in a secure folder for further analysis.

## Image data extraction, and inter-rater reliability

The fluorescent image with the highest overall intensity was chosen from each experimental trial. A line perpendicular to the surface of the brain was drawn by two independent reviewers blinded from each other, attempting to find the trajectory of maximum diffusion in each trial Fig. 4a. These lines were created manually in ImageJ by heuristically starting at the surface of the brain, directing them perpendicular to the surface, along an axis where there is maximum fluorescence. A plot profile was created to measure the fluorescence intensity along the line (Fig. 4b). The maximum fluorescence was 255 relative fluorescence units (RFUs).

The plot profiles from both annotators were analyzed for inter-rater reliability. The mean correlation between the plot profiles was $0.98 \pm 0.02$ between both raters, with a slope of $1.05 \pm 0.20$. Discrepancies in slope were only observed in control trials, where infusion of the fluorophore was minimal. Given the high degree of agreement between annotators and the

high correlation between plot profiles, the plot profiles from one annotator were selected for analysis. The plot profiles extended up to 4 mm into the brain parenchyma, much further from the penetration infusion that could be observed. The minimum value in the plot profile, for each sample, was subtracted from the data so that the minimum value of the pre-processed data was always zero.

From the plot profiles, three metrics of interest were identified to characterize the effect of the experimental intervention compared to diffusion-only control. ECED is used to infuse a fluorophore from a doped hydrogel at the surface of the brain into the brain parenchyma. The distribution of the fluorophore infusion inside the brain parenchyma is of a skewed, long-tailed shape (Fig. 4b). Therefore, we analyzed two metrics to address the amount of fluorophore inside the brain: distance to 30% of the maximum intensity $d(FI_{30})$ in the plot and abbreviated as $d_{30}$, and distance to 10% of the maximum intensity $d(FI_{10})$ in the plot and abbreviated as $d_{10}$, starting from the brain surface. Additionally, we analyzed the displacement of the point of highest fluorescence intensity, measured by its displacement from the brain surface, labeled as $d_{Peak}$ in the schematic Fig. 4b. The fluorescence peak intensity, the maximum value of the plot profile, was another metric evaluated from each trial. This metric was analyzed as a percentage of the 255 RFU upper limit in each experiment.

The area covered by the fluorophore in the selected brain slice was analyzed to assess the amount of fluorophore in the slice. The area covered by the fluorophore at thresholds of 10% ($A_{10}$) of the maximum intensity and 30% ($A_{30}$) of the maximum intensity were taken. Pixels in the region of interest that contained fluorescence values over these thresholds (Fig. 3f, g, and Fig. 2f, g) were added to obtain the overall area covered by the fluorophore.

The maximum intensity is 255 RFUs at 200 ms exposure for ex vivo trials, and 400 ms exposure for in vivo trials. This exposure time was selected by evaluating all the images in each experiment. For ex vivo trials, the brain sample that had the maximum fluorescence intensity reached the 255 RFUs saturation level at 400 ms, while the brightest in vivo brain slice reached 255 RFUs at 200 ms. Therefore, the maximum fluorescence intensity used as the thresholds for $d_{10}$, $d_{30}$, $A_{10}$, and $A_{30}$ was the maximum intensity reached by at least one of the trials, separately for the ex vivo and in vivo experiments. This experimental design was not designed to compare ex vivo versus in vivo trials. Rather, the goal of the experiment design was to compare ex vivo ECED vs ex vivo diffusion-only control trials, and similarly for in vivo trials.

**Statistics and reproducibility**. In this study, $N$ represents the number of animals used in experiments. For quantification in brain slices, we used one slice per subject. All bar graphs are mean values ± error, representing the 95% confidence intervals. Statistical comparisons between two groups were made using unpaired, two-tailed, Welch's $t$-test for unequal variances. The statistical significance threshold was selected at $\alpha = 0.05$. In-text statistical results are reported as mean ± standard error of the mean (SEM). $t$-test results are presented as $t$(degrees of freedom) = [$t$-value], $p$ = [$p$-value]. Complimentary statistical tables of results for exposure times of 200 ms and 400 ms for both ex vivo and in vivo trials are presented in Table 1, Table 2, and Supplementary Table 1. Information from each experimental trial with outcome metrics is presented in Supplementary Data 1. All statistical analyses were performed using Matlab 2022a.

### Reporting summary
Further information on research design is available in the Nature Portfolio Reporting Summary linked to this article.

### Data availability
All data associated with this study are present in the article. The numerical source data for 200 ms and 400 ms exposure, for both in vivo and ex vivo trials, is found in Supplementary Data 1. The descriptive statistics for the 200 ms and 400 ms exposure data, for both in vivo and ex vivo trials, is found in Supplementary Table 1. All raw image data is present as a Dryad dataset[53],

doi:10.5061/dryad.m37pvmd78. All other data are available from the corresponding author on reasonable request.

### Code availability
The open-source packages used in our analysis are stated and cited in Materials and Methods, as Supplementary Code 1.

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

## Acknowledgements

The authors thank the members of the Department of Neurosurgery and our colleagues at Houston Methodist Research Institute, for helpful discussions throughout the project. Specifically, we would like to thank Allison M. Frazier, Cinzia Stigliano, Xiufeng Tang, Laura Montier, Aboud Tahanis, Hannah Flinn, and Morgan Holcomb for the fruitful discussion about the materials and the equipment used. We thank Nathanael Hernandez and Alessandro Grattoni for donating the ex vivo research subjects. Figure 1 was created in part with BioRender.com. This project was supported by the Houston Methodist Foundation and the Houston Methodist Research Institute Clinician-Scientist Award. The authors declare that this study received philanthropic funding from Paula and Rusty Walter and Walter Oil & Gas Corp Endowment at Houston Methodist. The funder was not involved in the study design, collection, analysis, interpretation of data, the writing of this article or the decision to submit it for publication. This study received philanthropic funding from the John S. "Steve" Dunn, Jr. & Dagmar Dunn Pickens Gipe Chair in Brain Tumor Research. The funder was not involved in the study design, collection, analysis, interpretation of data, the writing of this article or the decision to submit it for publication. This work was funded in part by grant number RP190587 from the Cancer Prevention and Research Initiative (CPRIT) and the Houston Methodist Foundation. L.S.B. is a Burroughs Wellcome Fund Scholar supported by a Burroughs Wellcome Fund Physician-Scientist Institutional Award to Texas A&M University of Physician Scientists. J.R.G. is a Burroughs Wellcome Fund Fellow supported by a Burroughs Wellcome Fund Physician-Scientist Institutional Award to Texas A&M University of Physician Scientists.

## Author contributions

J.G.C.G.: Methodology, software, validation, formal analysis, investigation, resources, data curation, visualization, writing original draft, writing review, and editing. L.S.B.: Methodology, software, validation, formal analysis, investigation, data curation, visualization, writing original draft, writing review, and editing. K.M.T.: Methodology, investigation, resources, writing

original draft. K.P.F.: Methodology, investigation. J.R.G.: Methodology, Investigation. M.K.H.: Methodology, investigation, resources, editing. F.H.: Methodology, investigation, resources. R.C.R.: Resources, project administration, supervision, funding acquisition. P.J.H.: Resources, project administration, funding acquisition. A.H.F.: Conceptualization, methodology, validation, project administration, supervision, funding acquisition.

## Competing interests

A.H.F. is an inventor in a patent for electroosmotic delivery assigned to University of Pittsburgh — of the Commonwealth System of Higher Education, Pittsburgh, PA (US). US Patent 11,471,674. Electroosmotic delivery was implemented in this study for fluorophore delivery to the brain. All other authors declare no competing interests.
