## [Peer review file · Communications Biology]

Reviewers' comments:

Reviewer #1 (Remarks to the Author):

In this study, Electrokinetic Convection-Enhanced Delivery (ECED) was investigated as a non-pressure-driven method to transport a neutrally charged fluorophore into the brain's local regions using an external electric field. Both ex vivo and in vivo experiments were conducted, demonstrating that ECED significantly increased the linear distance from the brain surface to 10% and 30% of maximum fluorescence intensity compared to control trials. Notably, ECED proved more effective in vivo, showcasing its potential for targeted solute delivery within the brain parenchyma, primarily through electroosmotic mechanisms. However, throughout the paper, there are still many insufficiencies in the experimental design, experimental logic, language description and data volume, etc. The comments and suggestions are as follows:

1. The rationale behind selecting the 10% and 30% thresholds for data analysis needs clarification. Their significance in the context of the study should be elaborated upon.
2. Biosafety considerations regarding the experimental group, control group, and the health of mice require further investigation. It is recommended to supplement the experimental data on biosafety, including factors like degradability, metabolic pathways, and survival cycles.
3. The method by which ECED distinguishes between normal brain tissue and lesions should be explained in detail.
4. In Figure 2-a, c, d, and e, aside from "changes in tissue porosity and structure in both in vivo and ex vivo brains," other potential reasons for the lack of significant variations in data, including the "Area of infusion value" and the "Fluorescence peak displacement value," should be addressed. Additionally, it is important to consider whether these non-significant experimental data influence the effectiveness of delivered fluorophores from ECED. Furthermore, it should be discussed whether the choice of 10% and 30% thresholds as analysis intervals impacts the non-differentiated experimental data. Exploring the impact of increasing the number of ex vivo samples on group differences is also recommended.
5. The analysis of variation between groups in Figure 3 may not be entirely valid when the sample sizes of the experimental group ($n = 6$) and the control group ($n = 7$) do not align.
6. The significant difference observed in the "Area of infusion" value between ex vivo and in vivo conditions in Figures 2 and 3 should be explained.
7. Consider reordering the presentation of data for better experimental coherence and economy. It is suggested that the sequence begins with the in vivo experiments followed by the ex vivo experiments.
8. Ensure consistency in the formatting of references and harmonize the reference style throughout the paper.

Reviewer #2 (Remarks to the Author):

This well-written manuscript describes a novel approach to the delivery of positively charged and neutral molecules to the brain compartment. The group uses Electrokinetic convection-enhanced delivery (ECED) which is based on electroosmosis from an acrylic acid/acrylamide hydrogel placed on the cortical surface. This method is designed to improve upon efforts that use more traditional convection-enhanced delivery to the brain. In the current form, the team uses the delivery capillary as one electrode and a second implanted electrode. Overall, this is a very strong paper with clear and compelling ex vivo data. In addition, in vivo data suggest better dye penetration in the case of the ECED group over convention-enhanced delivery. The work is designed as a proof-of-concept study and no functional improvement is demonstrated yet.

Important questions are:

- 1) What is the directionality of the delivery? How important is the positioning of the counter electrode?
- 2) How could the functional performance of this approach be assessed in the context of an application? This would greatly improve the potential impact.
- 3) How could this gel be used in the context of a surgical recession? This seems like a very compelling potential application but should be further evaluated.
- 4) The work is innovative and potentially addresses an important research topic. However, it appears premature in its current form and would benefit from an example demonstrating a functional advantage.

Responses to Referees

Date: February 13, 2024.

Dear Referees,

We very much appreciate the constructive feedback from your reviews, and we greatly appreciate your time. We have addressed each of the specific comments below in detail.

The reviewer comments are in **bold**, and our responses are in regular font.

Reviewers' comments:

Reviewer #1 (Remarks to the Author):

In this study, Electrokinetic Convection-Enhanced Delivery (ECED) was investigated as a non-pressure-driven method to transport a neutrally charged fluorophore into the brain's local regions using an external electric field. Both ex vivo and in vivo experiments were conducted, demonstrating that ECED significantly increased the linear distance from the brain surface to 10% and 30% of maximum fluorescence intensity compared to control trials. Notably, ECED proved more effective in vivo, showcasing its potential for targeted solute delivery within the brain parenchyma, primarily through electroosmotic mechanisms. However, throughout the paper, there are still many insufficiencies in the experimental design, experimental logic, language description and data volume, etc. The comments and suggestions are as follows:

1. The rationale behind selecting the 10% and 30% thresholds for data analysis needs clarification. Their significance in the context of the study should be elaborated upon.

We choose 10% and 30% thresholds because there was significant signal to noise issues after 10%. Once the data reached below the 10% threshold (25/255 fluorescence intensity scale), it was implausible to determine if the fluorescent microscopy data constituted actual signal or background noise. See Figures 2a and 3a for reference. The 10% and 30% thresholds have significant context because it demonstrates the distance drugs can penetrate before they reach 10-30% of the infused concentration, which is maximum at the initial infusion site. Using established pharmacokinetic data, such as minimum effective concentration and therapeutic index, drug concentrations can be adjusted so the region of interest is covered with effective drug concentrations (Faraji et al. 2020). This will be especially vital when infusing potentially toxic drugs which can damage surrounding healthy brain parenchyma.

The use of the 10% and 30% thresholds is arbitrary, intended to show the depth of penetration for a diluted (10%) and concentrated (30%) spread of the fluorophore (Line 602-613). In previous experiments (Faraji et al. 2020), a threshold chosen was 25% (line 340). Our decision for 10% in addition to the 30% threshold was meant to show different levels of infusion inside the brain.

We added further explanation in lines 485-489.

Faraji, Amir H., Andrea S. Jaquins-Gerstl, Alec C. Valenta, Yanguang Ou, and Stephen G. Weber. "Electrokinetic convection-enhanced delivery of solutes to the brain." ACS Chemical Neuroscience 11, no. 14 (2020): 2085-2093. <https://doi.org/10.1021/acscemneuro.0c00037>

Bodor, N., and Buchwald, P. (2008) Retrometabolic drugdesign: principles and recent developments. Pure Appl. Chem. 80,1669–1682.

Dinda, S. C., and Pattnaik, G. (2014) Nanobiotechnology-based Drug Delivery in Brain Targeting. Curr. Pharm. Biotechnol. 14,1264–1274.

2. Biosafety considerations regarding the experimental group, control group, and the health of mice require further investigation. It is recommended to supplement the experimental data on biosafety, including factors like degradability, metabolic pathways, and survival cycles.

We appreciate your response, and we acknowledge the importance of biosafety in any experimental conditions involving animal models. However, these experiments were completed without survival surgeries and fluorophores whose metabolic pathways are not completely determined. We do plan to complete future experiments with survival studies and using established drugs to determine both the feasibility and safety of ECED in both rat models and larger animal models.

3. The method by which ECED distinguishes between normal brain tissue and lesions should be explained in detail.

ECED does not distinguish between normal brain tissue and lesions. The electrodes are placed so drugs flow from the positive to negative cannula. The drugs are infused along current paths which depend on various factors including tissue architecture (Faraji et al., 2020). The electrodes can be adjusted so the current paths, and thus the directionality of flow, will minimally involve normal brain tissues and maximally involve the lesion of interest.

4. In Figure 2-a, c, d, and e, aside from "changes in tissue porosity and structure in both in vivo and ex vivo brains," other potential reasons for the lack of significant variations in data, including the "Area of infusion value" and the "Fluorescence peak displacement value," should be addressed. Additionally, it is important to consider whether these non-significant experimental data influence the effectiveness of delivered fluorophores from ECED. Furthermore, it should be discussed whether the choice of 10% and 30% thresholds as analysis intervals impacts the non-differentiated experimental data. Exploring the impact of increasing the number of ex vivo samples on group differences is also recommended.

The data is presented for in vivo and ex vivo samples separately to address this issue. We repeated the analysis by making one additional change: the data is now shifted by the minimum fluorescence intensity value, making the lowest value always zero. Line: 586-594.

The number of samples for ex vivo trials was expanded by one per group to evaluate the trend and better quantify the effect observed. Although the metrics of infusion distance, area of infusion, peak displacement, and peak maximum were not significantly different, we observed strong significant differences in the linear fluorescence intensity profiles from the surface the brain (Fig 2a). This effect, with the additional subjects included, and the pre-processing step added, demonstrate the effectiveness of the intervention in ex vivo as well. The non-significant experiment data further quantifies the observations: ECED seems to be less significant ex vivo compared to in vivo, mainly due to increased infusion in the control condition as well. We believe this to be an important observation that characterizes the effect of ECED as studied ex vivo.

5. The analysis of variation between groups in Figure 3 may not be entirely valid when the sample sizes of the experimental group (n = 6) and the control group (n = 7) do not align.

The authors do not believe that different number of samples per group is a limiting factor in the analysis, if the sample sizes are reasonable and the statistical method for comparison is chosen appropriately. However, we decided to move forward with the same number of samples per group in this manuscript revision.

After further inspection, one sample of the in vivo trials was inconsistent with the rest of the cohort: There was excessive background noise in the first of the control subjects, possibly due to contamination by PFA, at disperse locations away from the infusion site. This subject was removed from the analysis, yielding n = 6 per group. In addition, the new pre-processing step of subtracting the minimum intensity value, so that the minimum for each sample is zero, was implemented. The results did not change substantially, but the significance values are more conservative. We believe these changes in data quality reinforce the results obtained.

6. The significant difference observed in the "Area of infusion" value between ex vivo and in vivo conditions in Figures 2 and 3 should be explained.

Thank you for pointing out this item. Further explanation of the findings was included in lines 299-308.

7. Consider reordering the presentation of data for better experimental coherence and economy. It is suggested that the sequence begins with the in vivo experiments followed by the ex vivo experiments.

Thank you for your comment. We agreed and we have implemented this change accordingly.

8. Ensure consistency in the formatting of references and harmonize the reference style throughout the paper.

Thank you. We addressed this concerned and will work with the journal for further necessary additions.

Reviewer #2 (Remarks to the Author):

This well-written manuscript describes a novel approach to the delivery of positively charged and neutral molecules to the brain compartment. The group uses Electrokinetic convection-enhanced delivery (ECED) which is based on electroosmosis from an acrylic acid/acrylamide hydrogel placed on the cortical surface. This method is designed to improve upon efforts that use more traditional convection-enhanced delivery to the brain. In the current form, the team uses the delivery capillary as one electrode and a second implanted electrode. Overall, this is a very strong paper with clear and compelling ex vivo data. In addition, in vivo data suggest better dye penetration in the case of the ECED group over convention-enhanced delivery. The work is designed as a proof-of-concept study and no functional improvement is demonstrated yet.

Important questions are:

1) What is the directionality of the delivery? How important is the positioning of the counter electrode?

The directionality of delivery, as shown in several previous studies, occurs due to electroosmotic forces. The directionality is from the positive cannula towards the negative cannula for most drugs. Therefore, the positioning of the counter electrode is vital to determining the current path and therefore the direction of infusion (Faraji et al. 2020). We added an additional explanation in lines 405-409.

Faraji AH, Jaquins-Gerstl AS, Valenta AC, Weber SG. Electrokinetic infusions into hydrogels and brain tissue: Control of direction and magnitude of solute delivery. *J Neurosci Methods*. 2019 Jan 1;311:76-82. doi: 10.1016/j.jneumeth.2018.10.005.

2) How could the functional performance of this approach be assessed in the context of an application? This would greatly improve the potential impact.

We greatly appreciate the reviewer for bringing up this important point. Future studies are currently being planned to determine the functional performance including using a glioblastoma animal model and infusing chemotherapies using our method. ECED and traditional CED can be compared to establish functional advantage.

We described a potential application, for which we are currently preparing an experiment for, in lines 347-360. We additionally included a simple method to evaluate area coverage compared to currently-used methods of drug distribution in tumor resection cavities in lines 355-357.

3) How could this gel be used in the context of a surgical resection? This seems like a very compelling potential application but should be further evaluated.

This gel can be used for various applications. Previous studies have shown how a doped hydrogel placed on the cortical surface can be used to infuse various drugs (Basso J, Miranda A, Nunes S, Cova T, Sousa J, Vitorino C, Pais A. Hydrogel-Based Drug Delivery Nanosystems for the Treatment of Brain Tumors. *Gels*. 2018; 4(3):62. <https://doi.org/10.3390/gels4030062>). However, these studies relied on diffusion alone from the hydrogel, and ECED and enhance and increase the amount of infusion into the brain parenchyma.

Also, future studies are being completed to infuse doped gels into resection cavities. This can be used to deliver therapeutic models to resection margins after surgical resection.

We included an example for tumor resection cavities in lines 347-357.

4) The work is innovative and potentially addresses an important research topic. However, it appears premature in its current form and would benefit from an example demonstrating a functional advantage.

Thank you for your comments. This work is built on previous in vivo and ex vivo experiments and helps lay the foundations for future works, which will discuss the functional advantage. We will complete future experiments looking at functional advantage using tumor models and various drugs once this work is published.

REVIEWERS' COMMENTS:

Reviewer #1 (Remarks to the Author):

This revised manuscript has taken care all the concerns raised by this reviewer. Now is acceptable for publication.

Reviewer #2 (Remarks to the Author):

The proposal has been improved, but important comments about including an application or functional outcome have not been adequately addressed

Response to reviewers.

Communications Biology, submission: COMMSBIO-23-2176A

April 16, 2024

We thank the reviewers for their constructive feedback during this time. Below is a response to the reviewers' comments for submission: COMMSBIO-23-2176A.

Reviewers' comments:

Reviewer #1 (Remarks to the Author):

This revised manuscript has taken care all the concerns raised by this reviewer. Now is acceptable for publication.

Thank you for your feedback and your constructive comments.

Reviewer #2 (Remarks to the Author):

The proposal has been improved, but important comments about including an application or functional outcome have not been adequately addressed

Thank you for your constructive comments. We appreciate your feedback regarding the application of our proposed method of therapeutics delivery.

Line 388-399 in file "ECED_4_4_submission_v11_clean.docx":

We added a paragraph explaining current applications based on electro-osmosis, the basic physical mechanism of action for delivery explored in this article. We also outline further potential applications.